# The Effects of Nordic Walking with Poles with an Integrated Resistance Shock Absorber on Red Blood Cell Distribution and Cardiorespiratory Efficiency in Postmenopausal Women—A Randomized Controlled Trial

**DOI:** 10.3390/biology12020179

**Published:** 2023-01-23

**Authors:** Katarzyna Sobczak, Paweł Nowinka, Krystian Wochna, Katarzyna Domaszewska

**Affiliations:** 1Laboratory of Swimming and Water Lifesaving, Faculty of Sport Sciences, Poznan University of Physical Education, Królowej Jadwigi Street 27/39, 61-871 Poznań, Poland; 2Department of Cardiology-Pulmonology, Heliodor Swiecicki University Hospital, Poznan University of Medical Sciences, 61-701 Poznań, Poland; 3Department of Physiology and Biochemistry, Faculty of Health Sciences, Poznan University of Physical Education, Królowej Jadwigi Street 27/39, 61-871 Poznań, Poland

**Keywords:** menopause, red blood cell distribution, echocardiogram, spirometry, maximal oxygen uptake

## Abstract

**Simple Summary:**

During the ageing process, a number of changes in body systems and structures occur. Age-related reduction in exercise capacity is manifested by a rapid development of fatigue and reduced exercise activity. Independent authors have confirmed the adverse prognostic value of red blood cell distribution width (RDW-CV), both as an independent factor and in correlation with other parameters, in heart failure, coronary heart disease and myocardial infarction, and chronic obstructive pulmonary disease (COPD). Physical exercise, which can be used by people of all ages as a therapeutic method, improves and helps to maintain cardiorespiratory fitness and fatigue tolerance and is a means to maintain the health and functional performance of older people. Different types of physical exercise result in different post-training adaptations. There is increasing research on the positive impact of combined endurance and resistance training on physiological parameters and health. The aim of our study was to examine the impact of Nordic walking training with classic poles (NW) and NW training with poles with an integrated resistance shock absorber (NW with RSA) on RDW-CV levels and to assess correlations between RDW-CV levels and cardiorespiratory performance in postmenopausal women.

**Abstract:**

Background: Age-related reduction in exercise capacity is manifested by a rapid development of fatigue. Research confirmed the adverse prognostic value of red blood cell distribution width (RDW-CV), an independent factor in heart failure, coronary heart disease and myocardial infarction. Physical exercise improves and helps to maintain cardiorespiratory fitness. The aim of our study was to examine the impact of 8 weeks’ Nordic walking training with classic poles (NW) and NW training with poles with an integrated resistance shock absorber (NW with RSA) on RDW-CV levels and to assess correlations between RDW-CV levels and cardiorespiratory performance in postmenopausal women. Methods: In this study, 32 postmenopausal women (NW-16, NW with RSA-16) participated in eight weeks of walking training. The mean age of women was 66.56 ± 4.23 year. and BMI 26.99 ± 3.86 kg/m^2^. At the beginning and at the end of the study, spirometry and exercise tests were performed. Haematological parameters were determined in the venous blood. Results: Statistical analysis of differences in post-training changes in the parameters between the groups studied showed a significant difference in change in body weight (∆body weight) (*p* < 0.05; ES: 0.778), BMI (∆BMI) (*p* < 0.05; ES: 0.778), waist circumference (∆WC) (*p* < 0.05; ES: 1.225) and (∆RDW-SD) (*p* < 0.05; ES: 1.215). There were no changes in electrocardiographic and spirometric parameters. Conclusions: Based on the findings from the present study, it can be assumed that endurance and resistance exercise can significantly reduce disease severity and mortality. A clinical analysis of RDW levels, together with other cardiological and biochemical parameters, can provide practical prognostic information relating to cardiovascular disease, mortality risk and treatment outcomes.

## 1. Introduction

During the ageing process, a number of changes in body systems and structures occur. They result in progressive loss of physiological reserve, a process known as homeostenosis, which makes older people more susceptible to disease. The age-related reduction in exercise capacity is manifested by a rapid development of fatigue and reduced exercise activity. The impaired cardiorespiratory fitness observed in older people, including our study participants, undoubtedly results from age-related changes in left ventricular filling, left atrial hypertrophy, prolonged contraction and relaxation of the left ventricle and reduced inotropic, chronotropic and bathmotropic responses to β-adrenergic stimulation [1,2]. As regards the respiratory system, a reduction in forced expiratory volume in one second (FEV_1_), forced vital capacity (FVC) and respiratory muscle strength and an increase in residual volume (RV) can be observed. Moreover, partial oxygen pressure (PaO_2_) decreases (according to the following formula: 100 − (0.32 × age)) as a result of poor matching between pulmonary ventilation and alveolar perfusion [3,4]. The reduced oxygen consumption capacity observed in older people can also be a consequence of the age-related reduction in bone marrow capacity and anaemia of the elderly. Sarcopenia and reduced muscle mitochondrial capacity lead to impaired muscle strength and endurance. Reduced oxygen uptake and utilisation at rest and during physical activity results in reduced effectiveness of aerobic muscle metabolism and occurrence of the negative effects of anaerobic adenosine triphosphate (ATP) resynthesis [5,6]. The mechanism of hypoxia is of interest to many researchers due to its significant impact on the body’s metabolic processes and lifespan. Changes associated with the natural ageing process accumulate, leading to cellular and tissue dysfunction and failure. Age-related anatomical and physiological changes impair the functioning of the entire body.

Cardiovascular disease is the leading cause of death in Poland, accounting for 46% of all deaths. This is a similar rate to that seen in other European countries. The problem today is Poland’s high incidence of cardiovascular diseases—it is 50% higher compared to western European countries [7,8]. Prognostic indicators for poor outcomes in patients with heart disease include: low left ventricular ejection fraction (LVEF), tachycardia, hypotension, low maximal oxygen uptake (VO_2max_), age, ischaemic myocardial damage, high serum N-terminal pro-brain natriuretic peptide (NT–proBNP) levels, renal failure, diabetes and anaemia [9,10]. In addition, recent years have witnessed increasing focus placed on the prognostic value of RDW-CV in assessing morbidity risk [11,12]. The greater the anisocytosis (i.e., variation in the size of red blood cells), the higher the RDW-CV values. Independent authors have confirmed the adverse prognostic value of RDW-CV, both as an independent factor and in correlation with other parameters, in heart failure [13,14], coronary heart disease [15,16] and myocardial infarction [17,18]. RDW-CV has also been used to monitor patients with ischaemic stroke [19,20], chronic obstructive pulmonary disease [21,22] and renal failure [23,24]. RDW-CV values >14.5% have been found to be associated with poor prognosis in patients with these conditions. In clinical studies, elevated RDW-CV levels were found to be highly correlated with such factors as age and sex. Increased anisocytosis is more common in older people, especially women [23,25,26].

Numerous prospective and retrospective studies have shown that changes in the size of red blood cells have a major impact on the severity of cardiovascular diseases caused by changes in blood flow through blood vessels. The changes are more pronounced in obese patients with lipid disorders. Ananthaseshan et al. found in their study that changes in blood flow caused by erythrocyte anisocytosis lead to interactions between morphotic elements and the vascular endothelium, which results in overexpression of adhesive molecules and development of inflammation in the vascular wall [27]. Most clinical complications associated with high RDW-CV levels result from atherothrombotic events caused by platelet activation. Therefore, therapeutic weight loss, improvement of lipid profile and reduction of inflammation should reduce RDW-CV levels and thus improve cardiorespiratory fitness in older people, including our study participants [28].

Physical exercise, which can be used by people of all ages as a therapeutic method, improves and helps to maintain cardiorespiratory fitness and fatigue tolerance and is a means to maintain the health and functional performance of older people. Statistics show that by 2035, the proportion of people aged over 65 will have increased by approximately 10%, which will probably result in an increase in the social costs relating to the treatment and rehabilitation of older patients with chronic conditions. Therefore, maintaining the health and functional performance of older people is important [29,30]. It should be noted that a lack of physical activity is the fourth leading risk factor for mortality worldwide. Data show that approximately one-third of the global adult population are physically inactive [31,32]. There is increasing evidence for the importance of different forms of physical activity as effective geroprotective interventions. The intensity and type of exercise should be tailored to each individual’s health and exercise tolerance. Different types of physical exercise result in different post-training adaptations. Endurance training improves cardiorespiratory fitness, just as resistance training reduces blood pressure, improves lipid profile and helps to restore and maintain muscle mass and physical fitness in older age. There is increasing research on the positive impact of combined endurance and resistance training on physiological parameters and health [33,34].

One of the new forms of training is NW with RSA, which combines aerobic and strength training. An elastic tape between two permanent elements in poles with a resistance shock absorber (RSA) provides additional resistance by increasing the overall intensity of exercise. In their study, Marciniak et al. showed that compared with NW training with traditional poles, NW training with RSA poles improves muscle strength and endurance through increased activity of the muscles directly involved in the exercise and an approximately 20% higher exercise oxygen consumption [35,36].

The aim of the study was to examine the impact of NW training with classic poles (NW) and NW training with poles with an integrated resistance shock absorber (NW with RSA) on RDW-CV levels and to assess correlations between RDW-CV levels and cardiorespiratory performance, echocardiography and spirometry parameters in postmenopausal women. A literature review indicates that this is the first study to examine such correlations.

## 2. Materials and Methods

### 2.1. Participants

Initially, 50 women aged 60–75 were recruited for the study based on their medical history and cardiology tests. The cardiological assessment involved taking the candidates’ medical history, including history of cardiovascular disease and medication, blood pressure measurement, a 12-lead electrocardiogram and echocardiogram examination. The inclusion criteria for the study project were normal blood pressure, BMI < 30 kg/m^2^, age over 60 years, and postmenopausal period. The following exclusion criteria were applied (presence of at least one of the factors listed below): diseases of the locomotor system preventing independent movement, morbid obesity, active or post cancerous disease (ongoing radiation or chemotherapy treatment), liver diseases (ALT > 3× borderline), chronic kidney disease (eGFR < 30 mL/1.73 m^2^/min), acute inflammation (CRP > 5 mg/dL), unstable ischaemic heart disease, after an ischaemic or haemorrhagic stroke (<6 months), post-STEMI (ST-elevation myocardial infarction) women with a drug-eluting stent implantation, NSTEMI (non-ST-elevation myocardial infarction) (<12 months), respiratory diseases (chronic obstructive pulmonary disease (COPD), pulmonary hypertension), inherited metabolic disorders (phenylketonuria and galactosaemia), autoimmune diseases (celiac disease, systemic connective tissue disease, haemolytic anaemia, vitiligo, Addison’s disease, hyperbilirubinaemia), non-specific enteritis (Crohn’s disease and ulcerative colitis), psychological disorders, antibiotic therapy, steroid therapy (ongoing), drug and alcohol addiction (a daily consumption of more than 1 portion of alcohol). The research project was a randomized controlled trial (RCT study). After the initial qualification (*n* = 10), women were resigned from participation in the research project. On the first date, 40 women turned up for the study. Randomisation was performed as simple random allocation; each subject’s identifier was forwarded to a person who was not involved in the conduct of the study, and who performed blinded randomisation using a computer list. The researchers asked participants to maintain their current diet and physical activity levels. On the second date, 32 women (NW-16, NW with RSA-16) came forward for testing. Five women did not participate in the required number of training sessions (<80%), and three women started participating in additional physical activity during the study design period.

The Institute for Research in Biomedicine (IRB) at the University of Poznan Medical School has given its approval for the study (7 February 2019; Ethics Approval Number: 245/19). The study was conducted according to the Declaration of Helsinki and the National Statement and Human Research Ethics Guidelines. Respondents were informed of the details of the research programme and the possibility of opting out at any stage. Written consent was obtained from each subject.

### 2.2. Anthropometric Measurements

All measured parameters were evaluated both at baseline and after exercise intervention. A standard measuring technique was used to take anthropometric measurements; height was measured with an anthropometer (accuracy ± 1 mm) and body weight was measured with a digital scale (±100 g) with the use of the WPT 60/150 OW medical scales (Radwag^®^, Radom, Poland). BMI and body fat mass (FAT) were determined using the bioimpedance method (BIA) (TANITA MC-980MA, Tokyo TANITA, Japan). In the analysis, the manufacturer’s suggestions regarding accuracy and correctness of measurements were applied. Waist circumference was measured horizontally using a tape measure with a measurement accuracy of 1 cm.

### 2.3. Exercise Test

The surveys were conducted between the 13 of February 2019 and the 17 of April 2019 in the morning. The exercise test was performed in a certified exercise laboratory 2 h after breakfast. The breakfast was the same for all female participants in terms of calories and composition. Maximal oxygen uptake was assessed using the modified Astrand-Rhyming protocol with the use of the Kettler DX1 Pro ergometer (Ense-Parsit, Germany), whereas heart rate (HR) was monitored using the Polar A-5 pulse meter (Polar Electro Oy, Kernpele, Finland) [37]. The predicted VO_2max_ was read from the nomogram (Astrand 1954) or accompanying tables and multiplied by the Astrand and von Dobeln age correction factors [38]. 

### 2.4. Pulmonary Function Test

The pulmonary function was carried out by conventional spirometry using a spirometer (Cosmed, Rome, Italy). Direct evaluation was performed for lung volumes, capacities and flows through the procedures of Slow Vital Capacity (SVC) and Forced Vital Capacity (FVC) in accordance with the standards of the American Thoracic Society (ATS) and the European Respiratory Society (ERS), with the patient in a seated position [39]. Spirometric examinations were carried out twice, i.e., at the beginning of the follow-up and at the end of the follow-up. Results of vital capacity (VC), ratio of the forced expiratory volume in the first one second to the forced vital capacity of the lungs (FEV_1_/FVC), maximal expiratory flow at 75%, 50%, and 25% of the FVC (MEF_75_, MEF_50_ and MEF_25_) are shown as % of reference value [40].

### 2.5. Resting Transthoracic Echocardiogram

Echocardiogram examination was carried out using the General Electric VIVID T8 (General Electric Medical Systems, Vivid T8 Pro, HaifaIsrael) with a 1–4 MHz transducer. The study protocol included assessment of heart chambers and heart function in accordance with the recommendations of the European Association of Echocardiography [41]. The hemodynamic parameters for heart valve flow were measured using continuous and pulsed-wave Doppler. In addition, septal and lateral mitral annular velocities were measured in an apical four-chamber view using tissue Doppler imaging. The following parameters were determined: peak A: late diastolic mitral inflow velocity (A), peak E: early diastolic mitral inflow velocity (E), and peak E′: early diastolic mitral annular velocity (E’). The E/E′ and E/A ratios were then calculated using the values obtained. Blood pressure at baseline and 8 weeks was measured using an automated device by a cardiologist.

### 2.6. Morphological Blood Test

On each of the test dates, 10 mL of venous blood samples were taken from the ulnar vein at rest (fast for 12 h before a blood test; 7:00 a.m.) using an S-Monovette syringe (Sarstedt, Nümbrecht, Germany). Haemoglobin (HGB) concentration, haematocrit (HCT) value, total erythrocyte count (RBC), leukocyte count (WBC), red cell distribution width-coefficient of variation (RDW-CV) were measured immediately after blood collection and the samples were analysed with the use of the MYTHIC 18^®^ haematology analyser (PZ Cormay SA, Łomianki, Poland).

### 2.7. Training Programme

The training program was created based on the American College of Sports Medicine (ACSM) guidelines for older people in good health condition and lasted 8 weeks (16 training sessions, twice a week). Women were assigned to two groups based on the type of poles used: classic poles (NW group) and RSA poles (RSA group). The RSA group used poles with an integrated RSA with an elastic resistance of 4 kg (Slimline Bungy Pump, Sports Progress International AB, Västernorrland, Sweden). Both groups of women, participated in training sessions at the same time under the supervision of a qualified physiotherapist certified to conduct gymnastics for the elderly with appropriate qualifications from the International Nordic Walking Association. Each training session began with a warm-up that lasted 10–15 min. Then, the women performed walking training over a distance of approximately 4 km. After each half of the planned distance (approximately 1.7–2.2 km, at a pace of approximately 1 km per 10 min), participants performed strength exercises and balance training (15 min). During the training program, the walking distance gradually increased, from 3.5 to 4.5 km with speed of approximately 1.5 km/15 min. Exercise intensity was measured using a heart rate monitor (Polar Electro Oy, Kernpele, Finland) and increased from 50% to 70% HRR. The training took place in a city park; the subjects walked along the inner lanes of the park, on varied ground. The training programme has been described in two earlier publications [35,36].

### 2.8. Statistical Analysis

Group size was calculated based on value of VO_2max_ results from the Madden et al. publication, which is methodologically similar to our project [42]. After calculations using a power (1-beta probability of error) of 95%, an effect size of 0.90 and an alpha error of 0.05 (two-sided), seven female participants were allocated to each group (four NW and four NW with RSA), ensuring equal allocation between groups. Therefore, the study started with 32 women randomly divided between the NW (*n* = 16) and NW with RSA (*n* = 16) groups. The Shapiro-Wilk test was used to calculate the normality of the data distribution. For variables with non-normal distributions, the Mann-Whitney U test and Wilcoxon test were used to assess the significance of differences between groups and study dates, respectively. Spearman rank analysis was used to calculate correlation coefficients. For significant changes, effect size (ES), according to Cohen’s criteria, an effect size ≥0.20 and <0.50 was considered small, an effect size ≥0.50 and <0.80 was considered medium and an effect size ≥0.80 was considered large [43]. Dell Statistica software (version 13, software.dell.com, Dell Inc., Round Rock, TX, USA) was used for calculations. Data are presented as means and standard deviations (SD). Statistical significance was set at *p* ≤ 0.05.

## 3. Results

Ultimately, the results of 32 patients were statistically analysed. The detailed anthropometric characteristics and maximal oxygen uptake of the participants are shown in Table 1. Statistical analysis of the anthropometric parameters studied, as measured on the first test date, showed a significant difference in body weight (*p* < 0.05; ES: 0.734) between the groups studied. 

NW training with RSA poles resulted in a significant change in body weight, BMI, waist circumference and VO_2max_ (*p* < 0.05), whereas in the case of NW training with classic poles, a significant change was only observed for VO_2max_ (*p* < 0.05). Statistical analysis of differences in post-training changes in the parameters analysed between the groups studied showed a significant difference in change in body weight (∆body weight) (*p* < 0.05; ES: 0.778), BMI (∆BMI) (*p* < 0.05; ES: 0.778) and waist circumference (∆WC) (*p* < 0.05; ES: 1.225).

The echocardiogram tests performed showed that none of the participants had significant valvular disease (Table 2). In all the participants, left ventricular contractility was over 50% and remained normal. An analysis of left ventricular diastolic function parameters, as measured with an echocardiogram, showed a statistically significant decrease in A-wave velocity (*p* < 0.05; ES: 0.352) and the E/A ratio (*p* < 0.05; ES: 0.301) in women participating in NW training with classic poles. A decrease in A-wave velocity and an increase in the E/A ratio are associated with deteriorating diastolic function and increasing filling pressure of the left ventricle. However, it would be unreasonable to conclude that the training resulted in a decrease in diastolic function in the participants. The tests were performed after an 8-week training programme and the changes in A-wave velocity may be temporary. Statistical analysis of post-training changes (∆) in the cardiological parameters analysed showed no significant differences between the groups.

As regards the haematological parameters analysed, a statistically significant difference was observed between the groups in RBC counts (*p* < 0.05; ES: 0.679) and RDW-CV values (*p* < 0.05; ES: 1.248), as measured on the first test date. The two forms of training used in the project resulted in a decrease in WBC counts (*p* < 0.05) and haematocrit values (*p* < 0.05). Moreover, in the case of the ‘NW with RSA’ group, a decrease in red blood cell count, red cell distribution width-coefficient of variation and red cell distribution width-standard deviation was observed (*p* < 0.05). The only significant difference between the groups in terms of post-training changes in the haematological parameters analysed was the difference in change in red cell distribution width-standard deviation (∆RDW-SD) (*p* < 0.05; ES: 1.215) (Table 3).

The spirometric assessment of the participants showed that none of them had reduced respiratory fitness. The groups were homogenous in terms of the variables analysed. Moreover, no statistically significant differences were observed between the groups in post-training changes (∆) in the parameters analysed. In the case of NW training with RSA poles, the only significant change observed in the parameters analysed was the significant increase in the FEV_1_/VC ratio (*p* < 0.05; ES: 0.679) (Table 4).

In the first period of the study (at the beginning of the training), a correlation was observed between RDW-CV and VO_2max_ (r = 0.4779, *p* < 0.05) and between RDW-SD and WC (r = 0.3691, *p* < 0.05). As regards spirometric parameters, relationships were found between the FEV_1_/VC ratio and body weight (r = 0.4524, *p* < 0.05), between the FEV_1_/VC ratio and BMI (r = 0.5048, *p* < 0.05), between MEF 50% and body weight (r = 0.5264, *p* < 0.05), between MEF 50% and BMI (r = 0.5113, *p* < 0.05) and between MEF 50% and WC (r = 0.4433, *p* < 0.05). An analysis of the results of the cardiological assessment of the participants showed a relationship between E1 and body weight (r = −0.4834, *p* < 0.05), between E1 and BMI (r = −0.4263, *p* < 0.05), between E1 and WC (r = −0.4772, *p* < 0.05) and between the E/E′ ratio and WC (r = 0.3953, *p* < 0.05). An analysis of post-training changes within the groups studied showed a relationship between change in VO_2max_ (∆VO_2max_) and change in the E/E′ ratio (∆E/E′) (r = −0.5750, *p* < 0.05) and between change in A (∆A) and change in E′ (∆E′) (r = 0.5438, *p* < 0.05).

## 4. Discussion

In this randomised controlled study, we analysed the impact of two types of NW training, namely, NW training with classic poles and NW training with poles with an integrated resistance shock absorber (RSA), on RDW levels in postmenopausal women. We also analysed changes in the cardiorespiratory performance and aerobic capacity of the women studied. We found that both types of NW training were effective in improving aerobic fitness. However, NW training with RSA poles was more effective in terms of changes in such parameters as body weight, BMI, WC and the FEV_1_/VC ratio and in terms of reducing the level of erythrocyte anisocytosis.

RDW-CV is a morphotic parameter that is commonly measured as part of a routine clinical examination. It can provide prognostic information for cardiovascular disease at no additional cost. Findings from studies relating to the impact of physical exercise on reduction in RDW-CV levels can be equivocal. This is because anisocytosis can be caused by inflammation, oxidative stress or impaired iron absorption [12,28]. In older age, decline in physiological reserve and sarcopenia lead to reduced muscle regeneration capacity and chronic inflammation. Regular physical activity reduces inflammation and limits anisocytosis, thus reducing the risk of cardiac events caused by abnormalities in erythrocyte size [44]. Veeranna et al. studies have shown that a decrease in RDW-CV levels is associated with reduced severity of coronary heart disease and mortality in heart patients, independently of the levels of biochemical inflammation markers (CRP) [45].

The results of the present study showed a statistically significant decrease (by 3%) in RDW-CV levels in the group participating in NW training with RSA poles (*p* < 0.05; ES: 0.665), which probably resulted from the significant reduction in body weight (*p* < 0.05; ES: 0.981) and excessive abdominal adiposity (*p* < 0.06; ES: 0.322) observed following the endurance and resistance training programme. Moreover, we found a relationship between RDW-SD values and WC (r = 0.3691, *p* < 0.05) in the group of women studied. As the present study is the first to look into the impact of this relatively new form of physical activity on RDW-CV levels in older people, further research is needed to confirm our findings. NW training with lower resistance did not result in a significant change in the above-referred variables. Our findings concerning the mechanism of post-training changes in RDW-CV values are consistent with those of a study by Rondanelli et al., who found a relationship between the incidence of coronary heart disease and increased body weight (and in particular an increased waist circumference), elevated lipid levels and increased RDW-CV values [46]. In the present study, we found a similar relationship between RDW-SD levels and WC (r = 0.3691, *p* < 0.05) in the women studied. Mota et al. found an inverse relationship between the amount of daily endurance and resistance physical activity and RDW-CV values and the incidence of cardiovascular disease caused by anisocytosis [47]. A population study carried out by Loprinzi and Loenneke in 2015 based on data provided from a national sample of the adult population showed that participation in the recommended weekly amount of resistance physical activity is linked to 11 percent reduced odds of having increased RDW levels (*p* = 0.006) [48]. In the study referred to above, no controlled training programme was used, and the association found was based on self-reported levels of physical activity. The authors of the study noted that the association is independent of inflammatory status, as measured by CRP. One other mechanism proposed by researchers is oxidative stress, which reduces erythrocyte survival, leading to the premature release of erythrocyte precursors into the bloodstream [49]. Another mechanism that can explain the impact of resistance training on RDW is erythropoietin resistance induced by muscular cytokines [50].

Our present study found a significant correlation between RDW-CV values and VO_2max_ (r = 0.4779, *p* < 0.05), as measured prior to the training programme, in the women studied. Both the forms of NW training used in the project resulted in a significant increase in VO_2max_ in the groups studied (*p* < 0.05). Findings from our earlier study showed an 86% greater increase in VO_2max_ in women participating in NW training with classic poles (large effect size) compared to women participating in NW training with RSA poles (medium effect size) [35]. However, a significant post-training decrease in RDW levels was observed only in the case of NW training with RSA poles. The observed association may contradict the thesis that high RDW levels are linked to limited exercise capacity. In their study on patients with heart failure, Van Craenenbroeck et al. found that higher RDW levels were independently associated with impaired exercise capacity [51]. The observed post-training increase in peakVO_2_ was correlated with a reduction in RDW levels. In order to elucidate the mechanism of the impact of RDW on exercise tolerance, it is necessary to establish cause and effect and to answer the question of whether an elevated RDW is a marker of impaired exercise tolerance, or whether it plays a pathophysiological role in impaired oxygen transport to muscle cells. Sugie et al. found in their study that RDW, which is known both as a marker of exercise intolerance in patients with CHF and as a strong and independent risk factor for mortality in the general population, is not an independent marker of peak VO_2_ in older people. The authors also found relationships between peak VO_2_ and different features of sarcopenia, frailty and cachexia, which may provide insight into the importance of peak VO_2_ and the rate of its decrease as strong predictors of mortality in the general population [52].

VO_2max_ is significantly determined by cardiovascular and respiratory fitness and muscular metabolic capacity and its increase following training is mainly observed in the case of endurance training with an intensity close to the individual’s anaerobic metabolic threshold. Not only does high-intensity training not increase VO_2max_, it is not suitable for older people [53]. Strength training significantly improves muscle mass and strength in older people, helping them to carry out basic activities, increasing their daily energy expenditure and improving their body composition. Standard strength training leads to hypertrophy and metabolic changes in muscles only, without an increase in VO_2max_.

Physical activity is known to improve the capacity of the circulatory system and muscular oxygen consumption. Studies have not found spectacular post-training changes in respiratory function in older individuals [35,36]. With age, the respiratory system loses its elasticity, chest mobility becomes limited and the ventilation-perfusion ratio worsens [4,54]. In young and healthy individuals, respiratory fitness is not a factor limiting exercise performance, whereas in the elderly, it can significantly reduce VO_2max_. A study by Khosravi et al. showed a positive relationship between FEV_1_ and DLCO and peakVO_2_ [34]. An interesting hypothesis was proposed by Powers et al., who noted that impaired pulmonary gas exchange may limit VO_2max_, but only in highly trained athletes who exhibit exercise-induced hypoxaemia [55]. In our present study, in which the effectiveness of two forms of NW training was evaluated, we found a significant increase in the FEV_1_/VC ratio in women participating in NW training with RSA poles (*p* < 0.05; ES: 0.679). The importance of combined endurance and resistance training was also discussed by Khosravi et al. [34]. They found in their study on a group of physically inactive women that an 8-week combined resistance and endurance training programme had a greater effect on VC, FVC and FEF 25–75% compared to endurance training and resistance training (*p* < 0.05). The study found no significant effect of resistance training, endurance training and combined endurance and resistance training on FEV_1_ and the FEV_1_/FVC ratio.

Only combined strength and endurance training improves cardiorespiratory fitness and muscle mass [56]. The present study found that as a result of the resistance aspect it involves, NW training with RSA poles stimulates muscle metabolism and thus improves cardiorespiratory fitness, significantly reduces body weight and results in a greater reduction in RDW levels compared to traditional NW training.

## 5. Conclusions

It can thus be concluded that this form of physical activity should be considered for postmenopausal women, and further research should focus on investigating the impact of NW training with RSA poles on healthy individuals of different ages and patients with cardiovascular diseases. Based on the findings from the present study, it can be assumed that endurance and resistance exercise performed as part of cardiological rehabilitation can significantly reduce disease severity and mortality. Thus, conclusions should be drawn cautiously and should be supported with future research. A clinical analysis of RDW levels, together with other cardiological and biochemical parameters, can provide practical prognostic information relating to cardiovascular disease, mortality risk and treatment outcomes. To the best of our knowledge, this is the first published report that compares the effectiveness of NW training with RSA poles with that of the traditional form of NW, considering the relationships between RDW levels and cardiorespiratory fitness and aerobic capacity in postmenopausal women. According to the greater effectiveness of NW training with RSA in postmenopausal women in reducing weight, waist circumference or lowering the RDW-C index, it should be recommended in activity planning for this specific group of people. In the future research to clarify the described relationships, the study should also include a group of men and expand with a group of people of different ages and health status. The absence of changes of the long-term effect in our study does not allow the authors to fully assess the effectiveness of any form of marching training in the prevention of cardiovascular disease.

## Figures and Tables

**Table 1 biology-12-00179-t001:** The anthropometric characteristics and maximal oxygen uptake of the groups subjected to the study.

	NW (*n* = 16)			NW with RSA (*n* = 16)		
	Baseline	9 Weeks	*p*-Value	Baseline	9 Weeks	*p*-Value
Age (year)	65.04 (4.01)			67.62 (4.29)		
Age of menopause (year)	51.06 (4.68)			49.25 (4.34)		
Body weight (kg)	66.61 (10.42)	67.23 (11.04)	0.3636	74.49 (11.18)	73.59 (10.95)	0.0083 (ES: 0.981)
Body height (cm)	160.93 (6.00)			162.19 (4.14)		
BMI (kg/m^2^)	25.68 (3.37)	25.92 (3.70)	0.3823	28.31 (3.96)	27.97 (3.36)	0.0124 (ES: 0.167)
FAT (%)	35.32 (4.51)	34.66 (4.07)	0.7563	37.97 (4.52)	37.85 (4.66)	0.5407
WC (cm)	82.66 (9.22)	81.67 (8.71)	0.0504	87.82 (9.53)	84.86 (8.82)	0.0052 (ES: 0.322)
VO_2max_ (mL/kg/min)	28.19 (4.79)	32.77 (5.04)	0.0011 (ES: 0.932)	28.20 (4.44)	30.57 (4.45)	0.0262 (ES: 0.533)

Data are presented as mean (SD), BMI—body mass index, FAT—body fat mass, WC—waist circumference, VO_2max_—maximal oxygen uptake.

**Table 2 biology-12-00179-t002:** The echocardiogram examination parameters and blood pressure of the groups subjected to the study.

	NW (*n* = 16)			NW with RSA (*n* = 16)		
	Baseline	9 Weeks	*p*-Value	Baseline	9 Weeks	*p*-Value
E (cm/s)	0.64 (0.13)	0.64 (0.10)	0.9499	0.69 (0.16)	0.64 (0.17)	0.3343
A (cm/s)	0.83 (0.16)	0.77 (0.18)	0.0144 (ES: 0.352)	0.81 (0.18)	0.78 (0.15)	0.2343
E/A	0.81 (0.29)	0.91 (0.37)	0.0328 (ES: 0.301)	0.89 (0.32)	0.83 (0.26)	0.4432
E′ (cm/s)	0.08 (0.02)	0.08 (0.03)	0.6121	0.08 (0.02)	0.08 (0.02)	0.1423
E/E′	8.09 (2.26)	8.09 (2.31)	0.8261	9.02 (2.26)	8.79 (2.65)	0.9547
SBP (mmHg)	127.81 (9.83)	126.67 (6.99)	0.5563	128.44 (8.11)	133.12 (9.29)	0.0559
DBP (mmHg)	77.50 (5.48)	78.00 (4.93)	0.6378	78.12 (5.12)	80.31 (2.87)	0.1925

Data are presented as mean (SD), A—peak A late diastolic mitral inflow velocity, E—peak E early diastolic mitral inflow velocity, E’—peak E’ early diastolic mitral annular velocity, SBP—systolic blood pressure, DBP—diastolic blood pressure.

**Table 3 biology-12-00179-t003:** Basic characteristics of haematological parameters of women subjected to an eight-week NW and NW with RSA training programme.

	NW (*n* = 16)			NW with RSA (*n* = 16)		
	Baseline	9 Weeks	*p*-Value	Baseline	9 Weeks	*p*-Value
WBC (10^9^/L)	6.46 (1.43)	5.56 (1.26)	0.0002 (ES: 0.668)	6.09 (1.09)	5.54 (1.15)	0.0199 (ES: 0.491)
RBC (10^12^/L)	4.37 (0.23)	4.34 (0.24)	0.5321	4.60 (0.42)	4.51 (0.42)	0.0121 (ES: 0.214)
HGB (mmol/L)	8.52 (0.39)	8.59 (0.36)	0.3635	8.64 (0.57)	8.59 (0.61)	0.2213
HCT (%)	39.34 (1.80)	38.41 (1.66)	0.0309 (ES: 0.537)	39.94 (2.69)	38.51 (2.88)	0.0019 (ES: 0.513)
RDW-CV (%)	12.41 (0.54)	12.33 (0.47)	0.7174	13.09 (0.55)	12.70 (0.62)	0.0146 (ES: 0.665)
RDW-SD (fl)	40.87 (1.77)	39.78 (2.18)	0.0787	41.14 (1.61)	38.46 (1.69)	0.0008 (ES: 1.624)

Data are presented as mean (SD), WBC—white blood cell count, RBC—red blood cell count, HGB—haemoglobin concentration, HCT—haematocrit value, RDW-CV—coefficient of variation red blood cell distribution width, RDW-SD—standard deviation of red blood cell distribution width.

**Table 4 biology-12-00179-t004:** Spirometry parameters of women subjected to an eight-week NW and NW with RSA training programme.

	NW (*n* = 16)			NW with RSA (*n* = 16)		
	Baseline	9 Weeks	*p*-Value	Baseline	9 Weeks	*p*-Value
%VC (%)	102.88 (18.43)	109.91 (19.75)	0.8509	112.97 (12.92)	115.92 (16.44)	0.0619
FEV_1_/VC (%)	106.67 (11.87)	110.04 (6.77)	0.1961	105.51 (8.59)	110.81 (6.67)	0.0019 (ES: 0.679)
%MFE 75 (%)	92.24 (23.37)	93.42 (27.03)	0.9749	94.24 (21.64)	93.71 (19.28)	0.6949
%MFE 50 (%)	96.12 (27.08)	96.85 (26.38)	0.7298	93.87 (30.67)	93.25 (26.43)	0.6949
%MFE 25 (%)	98.06 (38.15)	107.79 (32.81)	0.1401	96.31 (28.16)	106.83 (30.72)	0.0843

Data are presented as mean (SD), VC—vital capacity, FEV_1_—forced expiratory volume in one second, MFE 75, 50, 25—Maximal Expiratory Flow.

## Data Availability

The data presented in this study are available on request from the corresponding author. The data are not publicly available due to the consent provided by participants on the use of confidential data.

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
