# Peer review of "The Effects of Nordic Walking with Poles with an Integrated Resistance Shock Absorber on Red Blood Cell Distribution and Cardiorespiratory Efficiency in Postmenopausal Women—A Randomized Controlled Trial"

_biology, 2023, doi:10.3390/biology12020179_

Round 1

Reviewer 1 Report

Too small research group. Could be more numerous.
Body composition measurements are missing. Excessive heterogeneity in body weight between groups.
The training time of 8 weeks seems too short. I suggest adding a limitation paragraph.

Author Response

Dear Reviewer

 Thank you for providing these insights. We wish to express our sincerest appreciation for your insightful comments on our paper, which have helped us significantly improve the quality of our paper. Below, we address each comment and indicate the location of changes, which are marked in yellow, in the revised manuscript.

Best regards,

Authors

Reviewer 2 Report

Dear,

Manuscript Number: biology- 2116680

Title Manuscript: The Effects of Nordic Walking With Poles With an Integrated Resistance Shock Absorber on Red Blood Cell Distribution and Cardiorespiratory Efficiency in Postmenopausal Women- A Randomized Controlled Trial

This RCT study investigated the effects of 8-9 weeks of two types of exercise methods including Nordic walking with classic poles and Nordic walking with poles-integrated resistance shock absorber on anthropometric and physiological characteristics, echocardiography parameters, hematological parameters, and spirometry parameters in 32 postmenopausal women aged 60-75 yrs. This study is an important and interesting topic since the study population is a group of postmenopausal women but at the moment MAJOR REVISIONS are necessary in order to make it suitable for a final decision for “Biology”.

POINTs of STRENGTH:

1) The effects of both Nordic walking with classic poles and Nordic walking with poles-integrated resistance shock absorber on physiological, echocardiography, hematological and spirometry parameters in postmenopausal women in a RCT study;    

POINTs of WEAKNESS (and/or should be revised to improve the manuscript):

Abstract:

2) To use an abbreviation, please write the full name in the first instance and follow it immediately by the abbreviated version in brackets in the manuscript text such as “RDW-CV” in the Abstract section and so on;

3) It is suggested to present a sentence regarding the importance of Nordic walking and cardiorespiratory function/fitness parameters in postmenopausal women in the Background section;

4) The exercise follow-up (8 weeks or 9 weeks) and type of exercise are not specified in the objective/aim section; please specify clearly;

5) 32 women? OR 32 postmenopausal women? Please modify in the methods section of the Abstract;

6) The mean age and BMI of participants as well as type of exercise are not specified in the methods section; please clarify;

7) Please provide the “Results section” based on the aims of the study such as between-group differences in physiological, echocardiography, hematological and spirometry parameters and as well as correlations among parameters;    

8) Please report the “conclusion section of the Abstract” based on the results obtained from the study;

9) Please add the keyword of “menopause” in the keywords section;

1. Introduction:

10) To use an abbreviation, please write the full name in the first instance and follow it immediately by the abbreviated version in brackets in the manuscript text such as “RDW-CV, ATP, NW, and RSA” in the introduction section and … ;

11) The hypothesis and purpose of this study can be stated in more detail, especially for physiological, echocardiography, hematological and spirometry parameters;

2. Materials and methods

2.1. Participants

12) Please report the type of study (a RCT study) in the methods section;

13) The recruitment process and screening of the study participants, especially inclusion and exclusion criteria should be described in more detail such as age, age of menopause, BMI, healthy status, blood pressure, free of medication, respiratory disorders, and so on;

14) The authors reported that “Initially, 50 women …… and …..Finally, 40 women were included in the project and were randomly assigned to two groups”. Why were 10 postmenopausal women excluded from the study? Please specify the reason with its number; in addition, were the participants 40 postmenopausal women or 32 postmenopausal women at baseline? Please clarify and modify;

2.2. Anthropometric measurements

2.3 Exercise test

15) It seems that measured parameters were evaluated both at baseline and after exercise intervention; IF YES, please report in the manuscript text;

16) Please add other anthropometric parameters of the participants such as age of menopause, blood pressure, body fat, and … in Table 1;

2.4 Pulmonary function test

17) The authors reported that “examinations were carried out two times during the course of the project (at the beginning and at the end of the nine-week NW and NW with RSA training program)”; Follow-up of exercise intervention was 8 weeks? OR 9 weeks? Which is true? Please correct and modify throughout the text of the manuscript as well as Tables;

18) Please add other spirometry parameters of the participants such as FVC, FEV1%, FEF25–75, FIV1, FIV1%, and MVV in Table 4;

2.5 Resting transthoracic echocardiogram

19) Please add other echocardiography parameters of the participants such as left ventricular end‑diastolic diameter, left ventricular end‑systolic diameter, left ventricular septum diastolic diameter, left ventricular posterior wall diastolic diameter, Aortic root diameter, left atrial area diameter, right ventricular diameter, right atrial area volume, mitral E-wave deceleration time, Aortic velocity time integral, especially end-systolic volume (ESV), end-diastolic volume (EDV), left ventricular ejection fraction (LVEF), and cardiac output (Q) in Table 2;

2.6 Preparation of blood samples for analysis

20) How much blood was taken for the blood sample? AND fasting duration/hours? Please clarify;

2.7 Training programme

21) As mentioned above, follow-up of study protocol was considered 8 weeks? OR 9 weeks? Please correct and modify throughout the text of the manuscript as well as Tables;

3. Results

22) Please correct the unit of VO2max “(ml/kg/m)” to (ml/kg/min);

23) Did the nutritional status of the participants be controlled during follow-up of 8-9 weeks? Please explain and provide nutritional parameters in the form of a Table;

24) There was no control group (group without exercise intervention) in this study; is this considered a limitation for this study? Please explain;

4. Discussion and 5. Conclusions

25) As mentioned above, the authors will agree that the limitations section has to be expanded in this study;

26) Please report the “conclusions” section based on the results obtained from the study, especially for postmenopausal women;

 27) What does this study add to the literature? Please explain and add in the conclusions section.

References

28) References section is not always in accordance with the authors' guidelines. In particular, please check No. 1, 19, 22, 25, 29, 35, 39, and 40 for validation.

Best Regards

14 December 2022

Author Response

(The authors gave the same response as above.)

Round 2

Reviewer 2 Report

Dear Authors,

Manuscript Number: biology- 2116680

Title Manuscript: The Effects of Nordic Walking With Poles With an Integrated Resistance Shock Absorber on Red Blood Cell Distribution and Cardiorespiratory Efficiency in Postmenopausal Women- A Randomized Controlled Trial

I am very grateful to the authors for their efforts.

In general, this manuscript has found suitable content after correcting major revisions, and the modified revisions are accepted.

Best Regards

19 January 2023